# A Mini Review on Osteoporosis: From Biology to Pharmacological Management of Bone Loss

**DOI:** 10.3390/jcm11216434

**Published:** 2022-10-30

**Authors:** Kok-Yong Chin, Ben Nett Ng, Muhd Khairik Imran Rostam, Nur Farah Dhaniyah Muhammad Fadzil, Vaishnavi Raman, Farzana Mohamed Yunus, Syed Alhafiz Syed Hashim, Sophia Ogechi Ekeuku

**Affiliations:** Department of Pharmacology, Faculty of Medicine, Universiti Kebangsaan Malaysia, Cheras, Kuala Lumpur 56000, Malaysia

**Keywords:** bone, menopause, osteoblast, osteoclast, osteocyte, skeleton

## Abstract

Osteoporosis refers to excessive bone loss as reflected by the deterioration of bone mass and microarchitecture, which compromises bone strength. It is a complex multifactorial endocrine disease. Its pathogenesis relies on the presence of several endogenous and exogenous risk factors, which skew the physiological bone remodelling to a more catabolic process that results in net bone loss. This review aims to provide an overview of osteoporosis from its biology, epidemiology and clinical aspects (detection and pharmacological management). The review will serve as an updated reference for readers to understand the basics of osteoporosis and take action to prevent and manage this disease.

## 1. Introduction

The world has been experiencing an increase in lifespan due to improved medical care and living environment, but this has not kept pace with the increase in healthy life expectancy [1]. Ageing causes multiple adverse physiological changes to the body due to the lifetime accumulation of molecular and cellular damage [2]. Among these geriatric diseases, the ageing of the skeleton is one aspect often overlooked by the community and medical professionals alike. Generally, humans achieve peak bone mass in the third decade of life, but the exact age varies with sex and skeletal sites [3]. After peaking, both sexes experience a decline in bone mass [4], which is accelerated during menopause in women [5]. 

Osteoporosis is a skeletal disease characterised by reduced bone strength due to deteriorating bone mass and bone microarchitecture, leading to increased susceptibility to fracture [6]. Owing to a lower peak bone mass and faster bone loss during menopause, women are at greater risk for osteoporosis than men [5]. While the development of osteoporosis is mostly asymptomatic, its ultimate consequences, i.e., fragility fractures, pose tremendous medical and economical challenges to the patients and society [7]. Despite the availability of effective therapy, a substantial number of patients with osteoporosis remain untreated [8].

In light of the importance of osteoporosis to the geriatric population, this review aims to provide an updated overview of osteoporosis, covering its epidemiology, pathophysiology, detection and pharmacological management. 

## 2. Pathophysiology of Osteoporosis

The traditional pathophysiological models of osteoporosis are based on endocrine mechanisms. Two examples are estrogen deficiency in postmenopausal women and secondary hyperparathyroidism in the elderly due to menopause and vitamin D deficiency. In reality, osteoporosis is a multifactorial disease caused by a complex interplay of genetic, intrinsic, exogenous, and lifestyle factors [9]. 

A basic understanding of the bone remodelling cycle will facilitate the discussion on the pathophysiology of osteoporosis (Figure 1). Osteoclasts, osteoblasts, and osteocytes are the three main players in bone remodelling. When bone damage occurs, the macrophage polykaryon-derived osteoclasts migrate to the damage site and perform bone resorption [10]. At the end of bone resorption, osteoclasts undergo apoptosis and produce apoptotic bodies that may play a role in the subsequent osteogenesis [10]. After the reversal phase, the mesenchymal stem cell-derived osteoblasts will migrate to the cavity and perform bone formation [11]. Some osteoblasts will be embedded in the bone matrix they synthesise and differentiate into osteocytes. Osteocytes act as a mechanosensor and play regulatory roles in regulating the bone remodelling process through signalling proteins and via perilacunar remodelling directly [12]. 

The bone remodelling process is coordinated delicately to maintain bone mineral homeostasis and strength. The differentiation of osteoclasts is stimulated by the receptor activator of the nuclear factor kappa-B (NF-kB) ligand (RANKL) and macrophage colony-stimulating factor (MSCF), and inhibited by osteoprotegerin (OPG) synthesised by osteoblasts and osteocytes [13]. The osteocytes synthesised sclerostin and Dickkopf-1 that inhibits the Wnt signalling pathway and osteogenesis by osteoblasts [14]. Bone loss occurs when the rate of bone formation is lower than bone resorption [15]. 

Many factors can influence the bone remodelling process, skewing it towards a catabolic direction. This review will focus on the discussion of the three most important contributors, which are estrogen deficiency, inflammation and oxidative stress. They serve as the basis of bone loss in many pathological conditions.

Estrogen deficiency due to menopause-associated cessation of ovarian function is a well-established cause of bone loss [16]. Recent studies showed that bone loss begins during the menopause transition due to the increase in circulating follicle-stimulating hormone [17]. Estrogen deficiency may even explain age-related bone loss in elderly men since androgens are converted to estrogens via the aromatase enzyme and exert bone protective effects [18,19]. The effects of estrogen deficiency have been replicated consistently in female castrated animals (Figure 2). The effects of estrogen deficiency on bone loss are mediated by the direct modulation of osteoblast, osteoclast, and osteocyte physiology via estrogen receptors on these cells. In particular, estrogen deficiency increases osteoclasts’ differentiation and survival, and causes the opposite effects on osteoblasts and osteocytes [20]. Estrogen deficiency is also linked to increased inflammation and oxidative stress, which further promotes bone loss [21]. Epidemiological studies have shown that estrogen deficiency is associated with increased pro-inflammatory cytokine production by peripheral blood mononuclear cells in women without comorbidities [22]. 

Recent studies have placed T cells (CD4+) in the central role of inflammation-induced osteoporosis. In particular, Th17 cells secrete several proinflammatory cytokines, such as interleukin (IL)-1, IL-6, IL-17, RANKL and tumour necrosis factor (TNF) and interferon (IFN)-γ, which are pro-osteoclastogenesis. Th17 also facilitates the secretion of RANKL by osteoblasts and osteocytes to support osteoclastogenesis [24]. Regulatory T (Treg) cells, which express transcription factor FOXP3 and are responsible for preventing excessive immune reactions and inflammation, have been shown to have an anti-osteoclastogenic role [25]. In a study, ovariectomised FoxP3-transgenic mice have been reported to be protected from bone loss, and the transfer of Treg to T cell-deficient RAG-1^−/−^ mice improves the bone mass of these mice [26]. These findings showed the importance of Treg in suppressing bone loss independent of other T-cells. Another study showed that IL-15 produced by dendritic cells is critical in activating the synthesis of IL-17A and TNF-α by memory T cells, and contributes to bone loss in ovariectomized mice [27]. Recent studies also revealed that a unique subset of CD4 + CD28− T-cells have higher pro-inflammatory and pro-osteoclastogenic properties than the usual CD28+ T-cells [28]. Apart from T cells, B cells abundantly found in the bone marrow are a significant source of OPG, RANKL and MCSF that regulate osteoclastogenesis [29]. 

Recent studies also have unveiled the relationship between the gut microbiome and bone health. Compared to normal individuals, patients with osteoporosis show an increased abundance of *Actinomyces*, *Eggerthella*, *Clostridium* Cluster XlVa and *Lactobacillus* [30]. The gut microbiome could regulate bone remodelling through several mechanisms, such as modulating the activation of lymphocytes and inflammation, influencing hormone and nitric oxide levels, altering the metabolism of vitamin D and calcium absorption, as well as regulating the intestinal-brain axis [31]. Of note, the immune system plays an important role in mediating the gut-bone axis. For instance, lymphocyte-deficient mice did not experience bone loss due to antibiotic-induced dysbiosis [32]. Gut dysbiosis also disrupts the synthesis of anti-inflammatory short-chain fatty acids such as butyrate [33]. Gut dysbiosis also increases intestinal permeability and circulating lipopolysaccharide (LPS) levels [34]. Apart from immune cells, LPS has been shown to stimulate the release of pro-inflammatory cytokines from osteoblasts and fibroblasts [35]. All these changes could induce bone loss. 

Oxidative stress represents another significant cause of osteoporosis. Free radicals are generated via aerobic respiration in the body. Under physiological conditions, the antioxidant system protects our body from the harmful effects of free radicals/oxidants. Oxidative stress is generated when the antioxidant system is overwhelmed by these oxidants, leading to the pathogenesis of various diseases, including osteoporosis [36]. The circulating level of endogenous antioxidants, such as uric acid and bilirubin, has been associated with bone mineral density (BMD) in large epidemiological studies [37,38]. Similarly, dietary or circulating antioxidant levels, for example, vitamin E and vitamin C, have been linked positively with BMD in human studies [39,40]. Experimental studies have shown that many risk behaviours for osteoporosis, such as cigarette smoking and alcohol consumption, are linked to increased oxidative stress [41,42]. These associations are contributed by the direct impact of oxidative stress on the physiology of bone cells. Oxidative stress is known to decrease the survival of osteoblasts and osteocytes and increase the differentiation of osteoclasts [43,44]. 

## 3. Detection and Diagnosis of Osteoporosis

Despite the notions of bone loss and skeletal micro-architectural deterioration mentioned in the Introduction, a more operational definition of osteoporosis is needed in the clinical setting. In 1994, the World Health Organization (WHO) put forward the operational definition of osteoporosis based on the T-score for BMD assessed by dual-energy X-ray absorptiometry (DXA) at the femoral neck or spine. This definition is still being used. An individual with a BMD value 2.5 standard deviations (SD) or more below the young female adult mean (T-score < −2.5) is categorised as having osteoporosis. A T-score between −2.5 and −1.0 is defined as osteopenia [45]. While the diagnosis of osteoporosis by the presence of a fragility fracture is universally accepted, measurement of BMD with DXA can accurately diagnose osteoporosis before a fracture occurs [46]. Several issues regarding this definition remain, like the “young adult reference” to be used in the calculation of T-score should be “young adult Caucasian women” or “sex and ethnic-matched young adults”. The selection of a different reference will influence the T-score [47,48]. Currently, the International Society for Clinical Densitometry supports the use of the reference from “young adult Caucasian women” for standardisation purposes [49].

Trabecular bone score (TBS) is a complementary index derived from the lumbar spine DXA scan, and it has received much attention in the last decade. It is a textural index that measures the variation of the grey pixels in the spine DXA image and estimates the 3D bone structure [50]. A high TBS indicates a better and stronger skeletal microarchitecture [50]. TBS has been found to predict fracture and provide additional information for fracture risk stratification [51]. Since it is generated from software from DXA images, no additional tools are required, and retrospective analysis of the images can be performed [52]. However, TBS does not escape from soft tissue interference and artefacts of spine DXA images. It also cannot be used alone to diagnose osteoporosis [52].

Quantitative ultrasound (QUS) is another bone health assessment technique. QUS assesses bone health by measuring the propagation of ultrasound waves (>20 kHz) at varying frequencies across bone sites of interest [53]. It is a convenient and rapid screening tool that could be used to prioritise patients for DXA scanning [54]. However, QUS cannot be used for the diagnosis of osteoporosis because the WHO’s cutoff is based on DXA, and the technology varies across different machines [55]. 

The opportunistic screening for osteoporosis can be performed using low-dose computed tomography (CT) imaging obtained for other clinical indications. Low-dose CT scans performed for other indications such as lung cancer screening can be used to assess volumetric BMD and screen for osteoporosis simultaneously with no extra equipment, patient time, or radiation exposure, and at no substantial additional cost [56]. This technique provides an opportunity to use CT scans for the diagnosis of osteoporosis and eventually for fracture risk assessment in countries where DXA is not widely available [57]. 

Other CT techniques commonly used for bone health determination include volumetric quantitative CT (vQCT) and high-resolution QCT (hsQCT). Trabecular BMD can be estimated with vQCT more precisely than single-slide QCT. It usually includes only two vertebrae to avoid the confounding effects of spinal artefacts, which can artificially increase the BMD reading. On the other hand, the QCT T-score is lower and incomparable with the DXA T-score, hence the classification of bone health using WHO’s criteria is not appropriate [58]. vQCT scan at the proximal femur is also possible, with a focus on cross-sectional areas at the neck and greater trochanter, as well as hip axis length [59]. The hsCT provides better trabecular and cortical morphology but requires a much higher radiation dose. However, the resolution is still borderline for direct trabecular microarchitecture determination. The results may vary significantly depending on the threshold and image processing techniques [59].

There are several risk factor-based algorithms available to screen for osteoporosis, e.g., Simple Calculated Osteoporosis Risk Estimation (SCORE), Osteoporosis Risk Assessment Instrument (ORAI), Osteoporosis Index of Risk (OSIRIS) and the Osteoporosis Self-Assessment Tool (OST) [60]. These tools are sensitive in detecting patients with low BMD (T-score < −2.5) but not so on osteoporotic fractures [61]. Using OST for Asians as an example, our group has demonstrated that ethic-specific adjustment of the cutoffs will improve the performance of these tools in detecting osteoporosis [62,63,64].

Several algorithms are available to be coupled to BMD measurement to assess the fracture risk of an individual. One of the most used algorithms is the FRAX^®^ developed by the University of Sheffield (www.shef.ac.uk/FRAX), which has been validated in 66 countries covering 80% of the world population [65]. It has been adopted in several country-specific guidelines for the initiation of treatment to reduce fractures [66]. FRAX may be used without BMD data, but the addition of BMD data improves the precision of fracture risk assessment [67]. 

Bone remodelling markers are circulating proteins released by the osteoblasts or osteoclasts or degradation products of the bone matrix that can be used to monitor the process of bone remodelling [68]. Some of the classic bone formation markers include bone-specific alkaline phosphatase, osteocalcin and N-terminal and C-terminal propeptides of type I collagen (PINP). Tartrate-resistant acid phosphatase 5b, C-terminal telopeptide of type I collagen (CTX), N-terminal telopeptide of type I collagen, pyridinoline and deoxypyridinoline are the commonly used bone resorption markers [69]. Bone remodelling markers are useful in monitoring the response of the patients to anti-osteoporosis therapy, but their utility in predicting osteoporosis and fragility fractures is limited due to the lack of standardization [70,71]. A meta-analysis has reported a modest but significant relationship between increased PINP and CTX and increased fracture risk [72], but their utility for individual risk prediction is still questionable. Others have investigated the use of novel bone markers from osteocytes (e.g., sclerostin and Dickkopf-1) in predicting osteoporosis and fracture, but without conclusive results [73,74]. The measurements of bone markers are affected by many preanalytical factors, including time and season of sampling, menstrual cycle, medical conditions, food intake and physical activity of the patients [69]. 

A surge in the studies of circulating microRNA (miRNA) as biomarkers for osteoporosis and fragility fracture has been noted recently. miRNAs are small non-coding RNA (~22 nucleotides) which bind with targets with complementary sequences and affect their translation [75]. A recent systematic review reported the dysregulation of miR-21 (primary -5p form), miR-125b, miR-100, miR-148a, miR-24, miR-328-3p, miR-124, miR-17, miR-152 and miR-335 in patients with osteoporosis compared to normal controls [76]. These miRNAs have been known to affect osteoblastogenesis and osteoclastogenesis [77]. A miRNA screening panel (osteomiRs) which measures 19 circulating miRNA has also been developed and has shown satisfactory performance in identifying women with osteoporosis defined by the WHO’s criteria (area under the curve = 0.830) or history of major osteoporotic fracture (area under the curve = 0.834) [78]. 

## 4. Epidemiology and Burden of Osteoporosis

Osteoporosis is a silent health threat to the elderly population worldwide, especially with the increase in longevity [79]. A recent meta-analysis in 2022 revealed that the prevalence of osteoporosis and osteopenia based on the WHO’s criteria was 19.7% [95% confidence interval (CI) 18.0–21.4%] and 40.4% (95% CI 36.9–43.8%) worldwide. The prevalence of osteoporosis was reported to be higher in developing countries (22.1%, 95% CI 20.1–24.1%) compared to developed countries (14.5%, 95% CI 11.5–17.5%) [80]. Ethnic differences in the prevalence of osteoporosis have been identified in the US, whereby African Americans had the lowest prevalence compared to Hispanics and Caucasians based on BMD of the femoral neck or lumbar spine [81]. Multiple studies on immigrants of Asian origin in the US revealed lower BMD among Asians compared to Caucasians [82,83]. However, the differences were attenuated when bone mineral apparent density was considered, showing that the difference could be due to bone size [82,83]. In Malaysia, with three major ethnic groups (Malays, Chinese and Indians), the differences in BMD and prevalence of osteoporosis were heterogeneous. In one study among women aged 45 years old and above, the hip BMD of the Chinese was significantly lower than Malays and Indians, but the difference was not apparent at the lumbar spine [84]. In another study, being Indians (as compared to Malays and Chinese) negatively predicted the presence of suboptimal bone health (T-score < −1) estimated from BMD of the hip/lumbar spine [85]. 

The prevalence of osteoporosis can also be estimated based on the incidence/prevalence of fragility fractures. These fractures frequently happen on the hip (proximal femur), spine (vertebrae) and wrist (distal forearm) after a low-energy impact [86]. In a study conducted in the European Union, Switzerland and the UK in 2019, 6.5 million men and 25.5 million women were found to have osteoporosis, with 4.3 million new fragility fractures [87]. In the Asian Osteoporosis Study, the age-adjusted hip fracture incidence for men and women per 100,000 was the highest in Singapore (164 and 442), followed by Thailand (114 and 289) and Malaysia (88 and 218) [88]. The trend was projected to increase in 2050 in both men and women, with a 3.55-fold increase in Malaysia, a 3.53-fold increase in Singapore, and a 2.79-fold increase in Thailand [56]. Interestingly, a study using the Medicare database in the US revealed a general downward trend of hip fracture incidence between 2000 and 2009, but the decline was the least among the Hispanic population, probably indicating a need for further osteoporosis prevention action [89]. In multi-ethnic Malaysia, the incidence of hip fracture was the lowest among Malays compared to Chinese and Indians [90]. This observation disagrees with the BMD findings, suggesting other factors influencing the ethnic difference in fracture risk. 

Fragility fracture carries significant healthcare and financial burden to the patients. A meta-analysis revealed that compared to their counterpart without fractures, women and men who sustained a hip fracture faced excess mortality of 8% and 18% after one year [91]. Fractures also negatively affected the quality of life of the patients by impairing mobility, self-care, and ambulation. The quality of life of the women after hip fracture also did not recover to pre-fracture levels after 10 years [92]. The cost for the investigations and management of osteoporosis has been a concern in the planning of health care policy. A meta-analysis reported that the pooled estimated hospitalization cost for patients with osteoporosis was $10,075, while the one-year health and social care costs were $43,669 [93]. In the UK, the cost of hip fracture is expected to increase from £ 3.5 billion/year in 2010 to £ 5.5 billion/year in 2025 [94]. In Malaysia, the cost of a hip fracture leading to hospitalization in 1997 was amounting to Ringgit Malaysia 22 million [90], whereas in Thailand the cost of treating a hip fracture was 116,458.6 Baht [95]. The cost of osteoporosis reflects the burden of osteoporosis, which most patients cannot afford. 

## 5. Risk Factors of Osteoporosis 

Risk factors for osteoporosis can be divided into both modifiable and non-modifiable ones, and both affect bone strength beginning in childhood (Table 1) [96]. Some of the non-modifiable risk factors for osteoporosis include age, female sex, ethnicity, and family history of osteoporosis. Age is a strong risk. As mentioned earlier, molecular and cellular damage accumulates with age, leading to the senescence of bone cells and imbalanced bone remodelling [2,97]. Furthermore, estrogen deprivation due to menopause in women and testosterone deficiency syndrome in men could compound the rate of bone loss [98]. The low-grade inflammation that occurs with ageing (inflamm-ageing) also promotes bone loss in elderly populations [99]. The pro-inflammatory environment causes skeletal stem/progenitor cell attrition, impairing bone regeneration [100]. 

Sex and ethnicity also play an important role in the development of osteoporosis in an individual. Women are at a significantly higher risk for osteoporosis. A recent meta-analysis reported the global prevalence of osteoporosis to be 23.1 (95% CI 19.8–26.9) among women and 11.7 (95% CI 9.6–14.1) among men [101]. As discussed earlier, this disparity is due to menopause and lower peak bone mass in women. To illustrate the influence of ethnicity on BMD, a cross-country comparison showed that Tobago Afro-Caribbean and African American women had higher hip BMD, whereas Hong Kong Chinese and Korean women had lower hip BMD compared to Caucasian women in the US [102]. Despite a lower BMD, total hip, spine, and wrist fracture incidence of Hong Kong Chinese was lower compared to Caucasian women in the UK. The difference was attenuated in men [103]. Thus, BMD alone may not explain the variation in fracture risk. Similarly, disparities in BMD and fracture incidence were also observed among ethnicity in multiracial countries like Malaysia [84,90]. 

A family history of osteoporosis is also an essential non-modifiable risk factor for the disease. A positive history of osteoporosis (usually among mothers) is a strong predictor of the same disease [104]. This is not surprising, since a 14-year longitudinal twin study among Caucasian women revealed that up to 56% of the individual variance in bone loss could be attributed to genes [105]. 

Since inflammation plays a very important role in promoting bone resorption, systemic inflammatory diseases like rheumatoid arthritis have been considered a risk factor for osteoporosis [106]. The increased circulating pro-inflammatory cytokines, the release of RANKL by T-cells and fibroblasts-like synoviocytes, and the presence of autoantibodies all contribute to the increase in osteoclastic bone resorption [107]. The use of certain medications, such as glucocorticoids, gonadotropin-releasing hormone antagonists, aromatase inhibitors, proton pump inhibitors, anticonvulsants, vitamin K-antagonists (warfarin) etc., are associated with increased bone loss [108,109,110,111]. Of these, glucocorticoid use is the most important cause of medication-induced osteoporosis due to its prevalent use and its potent effect in inhibiting osteogenesis through various direct and indirect mechanisms [112]. Warfarin use is a significant risk factor for bone loss because it affects the vitamin K-dependent protein carboxylation process [113,114]. The use of non-vitamin K antagonist oral anticoagulants is reported to be associated with a lower risk of fracture compared with warfarin users, especially with long-term usage [115].

Diabetes mellitus (DM) represents a unique risk factor for osteoporosis. In general, excessive bone marrow adipogenesis and the presence of advanced glycation products that are toxic to the bone cells in DM are bad for bone health [116]. However, the BMD phenotypes diverge between Type 1 and 2 DM [117,118,119]. Type 1 DM is associated with low BMD, probably due to the lack of bone anabolic signals from insulin; Type 2 DM is often associated with increased BMD due to increased circulating insulin and leptin, which provide anabolic signals to osteoblasts. Obesity often associated with type 2 DM also provides a mechanical stimulus for bone accrual. However, the pro-inflammatory and pro-oxidative stress environment in DM are not favourable to bone health [116,120,121]. The fracture risk of patients with type 2 DM remains high, suggesting the presence of non-BMD contributors, such as low bone strength and quality, diabetic ophthalmic and neurological complications and the effects of anti-diabetic medications [122].

On the other hand, modifiable risk factors of osteoporosis are preventable risk factors. Examples include smoking, physical inactivity, calcium intake, and alcohol consumption. Tobacco smoking can cause an imbalance in bone turnover, leading to lower bone mass and increased susceptibility to fracture. Apart from direct toxic effects on bone cells, chemicals in tobacco can cause alterations in calciotropic hormone levels and intestinal calcium absorption, dysregulation in sex hormone production and metabolism, adrenal cortical hormone levels and the RANK/RANKL/OPG system [42,123]. Currently, there are limited studies on the effects of e-cigarettes on bone health. However, given the toxicity of nicotine and other auxiliary chemicals from the vape, it might also pose a significant health risk to the skeletal system [124]. Another risk behaviour of osteoporosis is alcohol consumption. A meta-analysis revealed that individuals taking ≥2 drinks/day possess 1.63 relative risk ratio (95% CI 1.01–2.65) of developing osteoporosis [125]. Alcohol consumption depletes calcium reserves and damages the pancreas, leading to low vitamin D synthesis and poor calcium absorption. While moderate alcohol consumption has proven not harmful in some studies, chronic alcohol intake destroys bone mass and reduces bone development, causing the bones to be prone to fissure formation in humans [41].

Physical inactivity or a sedentary lifestyle is positively associated with osteoporosis development. An experimental study using the hindlimb unloading model showed that disuse leaves a significant impact on trabecular mass in the aged skeleton, and it is less responsive to the effects of mechanical reloading [126]. A recent review looking at the effectiveness of different exercises on bone health in patients with osteoporosis indicated that weight-bearing aerobic exercise may be effective in preventing progressive bone loss, and strength and resistance exercises may be effective in increasing muscle mass and BMD at specific sites. Multicomponent exercises may combine the benefits of both modes of physical activity, while the effectiveness of whole-body vibration on BMD remains controversial [127]. 

Calcium intake influences skeletal calcium retention during growth, thus affecting peak bone mass achieved in early adulthood [128]. Thus, calcium deficiency is another important risk factor for osteoporosis. Chronic calcium deficiency leads to hyperparathyroidism and mobilisation of skeletal calcium stores to maintain circulating ionised calcium levels [129]. The current recommended dietary calcium intake for normal adults is 800–1000 mg/day [130]. However, this level is not achieved in Asians due to their sensory aversion to dairy products, lactose intolerance and the cost of calcium-rich food [131]. A recent study among Malaysians aged 40 years and above showed that the average calcium intake was between 600–650 mg/day [132]. Vitamin D deficiency is another risk factor for osteoporosis, as it facilitates intestinal calcium absorption. A serum vitamin D level below 50 nmol/L will trigger a spike of parathyroid hormone, leading to skeletal calcium mobilisation [133]. Despite cutaneous vitamin D synthesis, vitamin D insufficiency is a problem in tropical regions such as Malaysia [134,135]. 

**Table 1 jcm-11-06434-t001:** Risk factors of osteoporosis.

Non-Modifiable Risk Factor	Explanation
Age	Accumulation of cellular damage, cellular senescence, chronic low-grade ageing and endocrine deficiency [100,136].
Female	Women have lower peak bone mass compared to men and accelerated bone loss during menopause [16].
Genetics	Polymorphisms of genes such as *VDR* [137], *ER* [138], *OPG* [139], *COL1A1* [140] and *TNFα* [141] are associated with osteoporosis and fracture risk.
Family history	A family history of fractures in parents, particularly at the hip, is significantly associated with fracture risk [142].
Medical conditions	Rheumatoid arthritis: characterised by systematic inflammation that is pro-osteoclastogenesis [107]. Hypogonadism: sex hormone deficiency that impairs bone formation and stimulates bone resorption [98] Chronic renal failure: characterised by dysregulation of mineral metabolism, leading to hyperphosphataemia, hyperparathyroidism, hypocalcaemia and decreased vitamin D synthesis [143]. Cancer: pathological cancer due to bone metastasis and the damaging effects of anticancer treatments on bone cells [144].
Medications	Glucocorticoids impair osteogenesis and indirectly stimulate effects on osteoclasts [112]. Sex hormone deprivation therapies induce sex hormone deficiency and subsequent bone loss [108]. Proton pump inhibitors induce mineral malabsorption and vitamin B deficiency [109] Anticonvulsants affect vitamin D conversation through cytochrome P450 enzymes, reduce calcitonin synthesis and calcium absorption [145]. Anticoagulants (vitamin K antagonists) prevent the action of bone Gla-protein and mineralisation process [114]. Non-vitamin K antagonist oral anticoagulants do not affect this process, thus not increasing fracture risk.
**Modifiable risk factors**	**Explanation**
Low body weight	Body weight exerts mechanical loading onto the bone [146]. Being underweight is also an indicator of malnutrition, which affects body metabolism [147]. Low body weight is a risk factor for fracture, and this relationship is associated with BMD [148].
Sedentary lifestyle	Physical activities exert mechanical loading onto the bone. They also increase the level of sex hormones and anti-inflammatory cytokines, as well as suppress pro-inflammatory cytokines in the system, which are protective of bone health [149].
Calcium deficiency Vitamin D deficiency	Calcium deficiency will induce hyperparathyroidism, which mobilizes calcium stores from the bone. The deficiency of vitamin D, which helps with calcium absorption, also causes the same effect [129].
Cigarette smoking	Chemicals in cigarettes are toxic to the bone cells, and induce changes in the RANK/RANKL/OPG, calcium absorption, sex hormones and cortisol levels, resulting in net bone loss [42].
Alcohol consumption	Alcohol reduces calcium reserves and damages the pancreas, leading to low vitamin D synthesis and poor calcium absorption [41].

## 6. Pharmacological Treatments for Osteoporosis

Pharmacological treatments for osteoporosis are divided into two types based on their mechanism of action, namely antiresorptive therapy, which decreases bone resorption; and anabolic therapy, which stimulates new bone formation [150]. Examples of antiresorptive agents are bisphosphates (BPs), denosumab, and selective estrogen receptor modulators. Teriparatide and the recently approved romosozumab are anabolic agents [151]. 

Bisphosphonates (BPs) are the first drugs introduced to reduce fracture risk among patients with osteoporosis. Alendronate, risedronate, ibandronate and zoledronate are examples of BPs. The BPs deposited onto the bone will inhibit farnesyl diphosphate synthase, an enzyme that regulates G-proteins controlling osteoclast function [152]. A meta-analysis demonstrated that BPs decreased overall fracture (odds ratio [OR] 0.62;), vertebral fracture (OR 0.55;) and non-vertebral fracture (OR 0.73;) in patients with osteoporosis [153]. However, prolonged use of BPs is associated with atypical femoral fracture and osteonecrosis of the jaw due to the severe suppression of bone remodelling. Other acute side effects of BPs include gastrointestinal distress, musculoskeletal pain, hypocalcaemia and ocular inflammation [154]. 

Denosumab is a monoclonal antibody that binds to and inhibits RANKL function, thereby reducing bone resorption activity and increasing BMD [155]. A meta-analysis of 11 trials showed that denosumab therapy significantly reduced the risk of clinical fractures [relative risk (RR) 0.57; 95% CI 0.51–0.63], nonvertebral fracture (RR 0.83; 95% CI 0.70–0.97), vertebral fracture (RR 0.32; 95% CI 0.25–0.40) and hip fracture (RR: 0.61; 95% CI 0.37–0.98) in patients with osteoporosis [156]. Common side effects of denosumab include back pain, skin rashes, lower extremity skin infections and pancreatitis. Injection site rashes can occur at administration sites [150].

Selective estrogen receptor modulators (SERMs) mimic the suppressive action of estrogen suppression on osteoclastic bone resorption. The first SERM approved for the management of postmenopausal osteoporosis is raloxifene. It improves lumbar vertebrae BMD by 2.5% after 2 years and decreases the relative risk of incident vertebral fractures by 30–50% in women with prevalent fractures or osteoporosis [157]. It also carries other non-skeletal benefits, such as improved lipid profile and breast cancer risk. However, it can cause venous thromboembolism and an increase in the risk of hot flashes and leg cramps [158]. The efficacy of SERMs is less than other agents, so they are recommended for women with osteoporosis/osteopenia but without a history of thromboembolism [157]. 

Teriparatide is an anabolic parathyroid hormone fragment which stimulates bone formation. According to a meta-analysis, teriparatide was more effective than BPs in preventing vertebral fractures (RR 0.55; 95% CI 0.40–0.77) and nonvertebral fractures (RR 0.65; 95% CI 0.46–0.90) [159]. It is also more effective than BPs in improving BMD in women with postmenopausal osteoporosis up to 18 months [160]. Common side effects of teriparatide include hypercalcaemia so routine monitoring of circulating calcium levels is necessary. Users might also experience nausea, vomiting and dizziness after taking teriparatide. Teriparatide is reported to cause osteosarcoma in rats, but this has not been observed in humans [161]. 

Romosozumab is a newly approved anabolic agent for osteoporosis management. It is a monoclonal antibody that binds sclerostin and increases bone formation [162]. In a recent meta-analysis of 10 randomised controlled trials, romosozumab is reported to increase BMD at the hip [mean difference (MD) 5.69; 95% CI 5.68–5.69], femoral neck (MD 5.18; 95% CI 5.18–5.19), and lumbar spine (MD 12.66; 95%CI 12.66–12.67) after 12 months. It is also associated with the reduced incidence of vertebral fractures [odds ratio (OR) 0.43; 95% CI 0.35–0.52] and nonvertebral fractures (OR 0.78; 95% CI 0.66–0.92) after 24 months [163]. It exhibits better safety profile efficacy in improving BMD than teriparatide [164]. However, a recent review of three phase III clinical trials indicated a numerical increase in cardiovascular events in patients with a cardiovascular history or at high cardiovascular risk [165]. Therefore, the cardiovascular profile of the patients should be considered when prescribing romosozumab. 

Sequential therapy involving anabolic agents and antiresorptive agents is being investigated since most patients with osteoporosis require long-term management. Patients with severe osteoporosis should start with anabolic followed by antiresorptive agents. However, the type of treatment, the severity of osteoporosis, and failure response will have to be considered to determine the treatment sequence [166].

## 7. Conclusions

Osteoporosis will continue to be a geriatric disease plaguing the world with an increasing elderly population. While ageing is an inevitable biological process, not every elderly person will have to experience osteoporosis. Cultivating osteoprotective behaviours at all stages of life will help to prevent excessive bone loss. Timely detection and treatment will prevent the occurrence of fractures and preserve the quality of life among the elderly. 

## Figures and Tables

**Figure 1 jcm-11-06434-f001:**
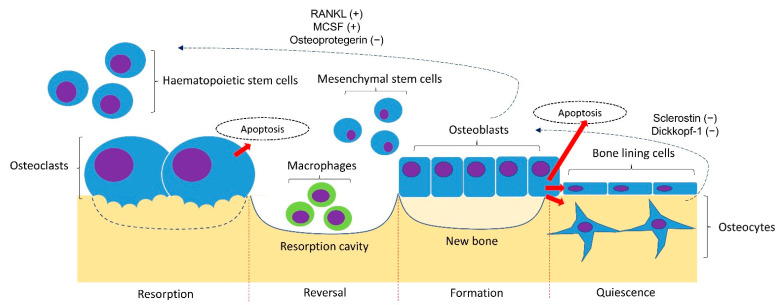
Bone remodelling cycle. The bone remodelling cycle is governed by osteoclasts, osteoblasts and osteocytes derived from the respective stem cell lineage. The differentiation of osteoclasts is stimulated by the receptor activator of nuclear factor kappa-B ligand (RANKL) and macrophage colony-stimulating factor (MSCF) and inhibited by osteoprotegerin (OPG) synthesised by osteoblasts and osteocytes. The osteogenesis of osteoblasts is inhibited by sclerostin and Dickkopf-1 synthesised by osteoblasts. Notes: +, promoting factor; −, inhibiting factor.

**Figure 2 jcm-11-06434-f002:**
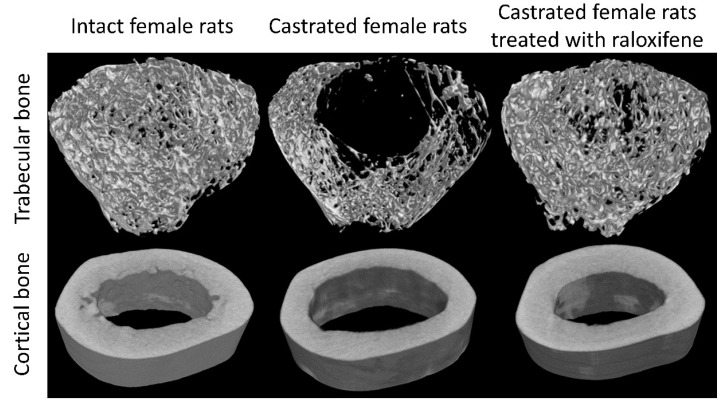
Skeletal microstructural changes of the rat’s femur following ovariectomy/castration and treatment with raloxifene (a selective estrogen receptor modulator). Significant deterioration of the trabecular bone and cortical bone can be observed with castration. Restoration of the microarchitecture is observed with raloxifene treatment. The images are modified from the article by Mohamed et al. [23]. Copyright owned by Kok-Yong Chin.

## Data Availability

Not applicable.

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
