# Peer review of "A Mini Review on Osteoporosis: From Biology to Pharmacological Management of Bone Loss"

_jcm, 2022, doi:10.3390/jcm11216434_

Round 1

Reviewer 1 Report

Overall, this article is well written. Some of my recommendations include:

-elaborating more on the detection and diagnosis regarding bone markers.

-elaborating more on the pharmacological treatment using romosozumab because it inhibits the wnt signaling pathway, especially when we talk about sclerostin. 

Author Response

Thank you for reviewing our manuscript. We are grateful for the constructive comments and have responded to each of them in the attached response sheet.

Reviewer 2 Report

The manuscript presented as a mini review is a thorough, concise summary of the findings in pathophysiology, clinic, diagnosis and therapy in osteoporosis at the time being. The review  adds to the special edition of our journal, giving not only an updated reference for readers but also insights into the latest scientific development of the mode of action of the disease .  However, I would like to comment on some of the topics discussed.

2. Introduction:

I am not sure whether osteoporosis can be defined as a degenerative disease, without mentioning the metabolic aspect in the definition since more and more younger people are affected.

The bone remodeling cycle is described very vividly and depicted in a didactive graph. The state of the art consideration of the  immunologic background and the role of proinflammatory cytokines is presented clearly.

The description of T (Treg) cells should also mention the possible role of the microbioma and bone turnover. The importance of oxidative stress is underlined properly.

3. Detection and diagnosis of osteoporosis:

Among the standard tools recommended by the WHO and IOF advantages and limitations of quantitative CT densitometry should be described, as well as high resolution peripheral CT densitometry and finally the add-on value of TBS.

The role of bone turnover markers should also be included in this chapter. In this context, I would like to encourage the authors to judge on status of certain micro RNA signatures.

4. Epidemiology and burden:

Epidemiologic data on the Asiatic region and comparison to other regions are very interesting and very important.

5. Risk factors:

In this chapter, diabetes and effects on bone should be described in more detail.

In the paragraph of drug-induced bone loss, the non-vitamin K-antagonist oral anticoagulants (NOAC) and their missing effect on bone and fracture risk should be mentioned.

6. Pharmacological treatment:

No further comments from my side.

Author Response

(The authors gave the same response as above.)
